# Reconfigurable shape-morphing dielectric elastomers using spatially varying electric fields

Ehsan Hajiesmaili [1] & David R. Clarke [1]

Exceptionally large strains can be produced in soft elastomers by the application of an electric field and the strains can be exploited for a variety of novel actuators, such as tunable lenses and tactile actuators. However, shape morphing with dielectric elastomers has not been possible since no generalizable method for changing their Gaussian curvature has been devised. Here it is shown that this fundamental limitation can be lifted by introducing internal, spatially varying electric fields through a layer-by-layer fabrication method incorporating shaped, carbon-nanotubes-based electrodes between thin elastomer sheets. To illustrate the potential of the method, voltage-tunable negative and positive Gaussian curvatures shapes are produced. Furthermore, by applying voltages to different sets of internal electrodes, the shapes can be re-configured. All the shape changes are reversible when the voltage is removed.

[1] John A. Paulson School of Engineering and Applied Sciences, Harvard University, 29 Oxford Street, Cambridge, MA 02138, USA. Correspondence and requests for materials should be addressed to D.R.C. (email: clarke@seas.harvard.edu)

The natural world abounds with shapes that morph as they grow, ranging from the evolution of ripples on leaves[1] to the complex folding of the human brain[2]. As their shapes change, geometrically their Gaussian curvature changes and, as understood from Gauss's theorema egregium, this is only possible because the deformations are spatially inhomogeneous[3]. Bending, homogenous expansion, or homogeneous contraction do not change the Gaussian curvature, defined as $\kappa = \kappa_1 \kappa_2 = \frac{1}{r_1 r_2}$, where $\kappa_1$ and $\kappa_2$ are the principal curvatures of the surface and $r_1$ and $r_2$ are radii of curvature. Therefore, under these deformations a body will not be able to morph from one shape to a fundamentally different one. To change the Gaussian curvature and morph in shape, gradients in deformation are required. One such mechanism that can give rise to shape changes is differential swelling[4]. A second mechanism is inhomogeneous stiffness[5–8]. A third mechanism is domain wall motion in magnetostrictive and ferroelastic materials[9].

Currently, actuators based on dielectric elastomers cannot morph in shape; there is no generalizable method for changing their Gaussian curvature. Current dielectric elastomer actuators are based on what might be termed a compliant capacitor model: a voltage applied to electrodes on opposite sides of a dielectric sheet creates opposite net charges. The Coulombic attraction between them produces a stress, the so-called Maxwell stress, that acts to thin the dielectric. For soft and incompressible materials, such as elastomers, the Maxwell stress induced thinning is compensated by expansion in the plane of the elastomer. These expansion strains can be large and used to drive an actuator[10,11]. However, the electric field remains homogeneous everywhere within the elastomer sheet and consequently, even though the elastomer is compliant, it cannot morph to a new shape; a flat sheet remains flat. Attaching a passive layer to the elastomer sheet results in bending actuators, for which the Gaussian curvature does not change either; although one principal curvature changes, the other principal curvature along the axis remains zero and therefore the Gaussian curvature remains zero. Actuators based on the compliant capacitor model either provide in-plane expansion[10], linear actuation[12,13], or bending[14]. There are also a few examples of dielectric elastomer actuators that can change their Gaussian curvature but using rigid frame. One example is a tunable lens[15] in which a sheet of dielectric elastomer is confined by a rigid frame and upon applying voltage the elastomer sheet increases its Gaussian curvature to accommodate the strain incompatibility. Another example is a prestretched sheet of elastomer supported by a flexible but stiff frame[16] which takes an initially negative Gaussian curvature and upon applying voltage the Gaussian curvature increases. Such methods for changing Gaussian curvature, however, are not generalizable to any more sophisticated shapes.

In this work, morphing of an initially flat elastomer sheet into shapes of different Gaussian curvature is demonstrated. To morph, according to Gauss's theorema egregium, a change of intrinsic curvature can only occur when the deformation is spatially inhomogeneous along the surface. In dielectric elastomers, this can be achieved by creating inhomogeneous electric fields when a voltage is applied. As will be shown, this can be accomplished by creating a multilayer structure consisting of a set of dielectric elastomer layers separated by compliant inter-digitated electrodes of different shapes. The meso-architecture of the internal electrodes defines the attainable shapes when voltages are applied. In contrast to the natural world, the shape changes are reversible when the applied voltages are removed. Furthermore, by altering the internal electric field distributions, the shape can be reconfigurable. Also, in contrast to other mechanisms for creating shape changes, the changes are not driven by temperature changes or diffusion of species or solutes and so are faster and amenable to electrical control and programing.

## Results

**Shape morphing through spatially varying electric fields.** The shape-morphing mechanism using inhomogeneous electric fields within a sheet of elastomer is exemplified in Fig. 1. It shows a stack of very thin circular, inter-digitated electrodes separated by thicker layers of circular elastomer sheets. The electrode diameters decrease from bottom to top. Upon applying a voltage between the ground and high-voltage electrodes, shown in gray and red in Fig. 1a, an electric field is induced inside the elastomer layers primarily in those regions where the two adjacent electrodes overlap (active regions, shown in red in Fig. 1b); everywhere outside these local regions, the electric field is negligible (passive regions, shown in blue in Fig. 1b). The electric field-induced deformation consists of a radial expansion of the elastomer between the overlapped electrode segments and a decrease in their spacing, which leads to a differential actuation: greater lateral expansion in the center of the elastomer sheet and decreasing towards the edge (shown by the thickness of the arrows in Fig. 1c), as well as greater expansion below the mid-surface. To accommodate the incompatible strain, the flat disk of elastomer morphs into a shape with positive Gaussian curvature, as shown in Fig. 1d using numerical simulations. The Gaussian curvature is a function of the applied voltage, position along the surface, and design of the electrodes.

Therefore, by designing the electrodes' geometry, one can define the active region within the elastomer sheet which determines both the metric tensor of the surface[4,17] and the applied bending moment. To create relatively simple shapes, as will be described, the appropriate electrode meso-architecture can be identified by a simple intuition and physical arguments aided by numerical analysis of actuation shape (the forward problem). More generally designing the meso-architecture of the electrodes, namely their geometry and arrangement, the spatial position of the actuating regions inside the elastomer and the differential actuation can be defined. More complex morphing shapes will require solution of the inverse problem.

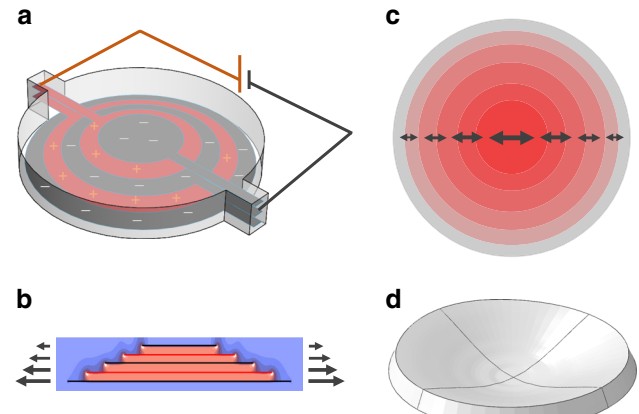

**Fig. 1** A generalizable method for shape morphing of thin sheets of elastomer by creating spatially varying internal electric field. **a** A multilayered structure of circular elastomer sheets interleaved with concentric, inter-digitated electrodes of decreasing radii with height. **b** The electric field is primarily concentrated in the regions of overlap between the adjacent electrodes as illustrated by the computed electric field distribution shown in orange. **c** Applying a voltage to the electrodes creates a radial actuation strain that varies with both radial position (represented by the length of arrows), and also with vertical position as shown by arrows in **b**. **d** In response, the circular disk deforms with a positive Gaussian curvature (simulation) that increases with increasing voltage

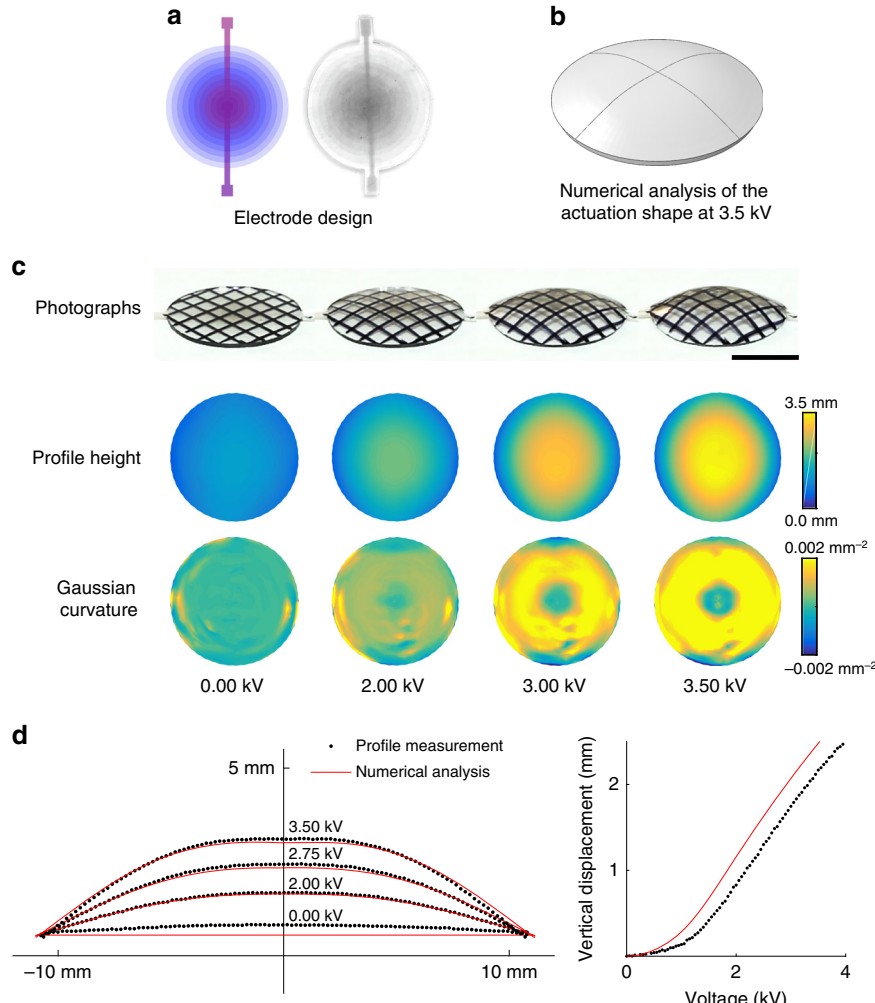

**Fig. 2** A thin sheet of dielectric elastomer morphing into a dome shape with positive Gaussian curvature. **a** The electrodes are designed such that the overlapping of the adjacent electrodes, which defines the active regions, decreases with the radius (shown schematically with color intensity on the left and a photograph on the right). **b** Numerical analysis of this structure shows that the thin sheet of elastomer with such electrode meso-architecture morphs into a dome, when a voltage is applied. **c** A series of photographs of the actuation shape at increasing voltages, together with the line scanner profile measurement data and the derived Gaussian curvature, show that the thin sheet of elastomer morphs into a dome with positive Gaussian curvature, increasing with the applied voltage. The elastomers are transparent and so a fiducial grid of black lines was marked on their surface for visualization. Scale bar: 10 mm.
**d** Comparing the simulation results (red line) and profiler measurement (black dots) for the cross-section of the actuation profile at different voltages (left) and the vertical displacement of the center of the disk as a function of the applied voltage (right) shows excellent agreement between the two

**Fabrication**. To test the shape-morphing concepts, dielectric elastomers were fabricated in a manner similar to those used for the manufacture of multilayer dielectric elastomer actuators[18]. Briefly, as described in greater detail in Methods, a mixture of acrylic oligomers, cross-linkers, and photo-initiators was spun coat onto a non-adhesive substrate and then ultraviolet (UV) cured in place. A mat of electrically percolating carbon nanotubes was then stamped through a mask, defining the shape and size of the electrode, onto the elastomer. Next, the mask was removed and the procedure repeated with different masks to produce the desired number of layers and the electrode meso-architecture. Finally, contacts were made to the individual electrode layers for actuation.

**Numerical analysis**. To complement the experimental demonstrations, as well as to aid in the design of the electrode meso-architectures, the actuated shapes as a function of applied voltage were computed using a fully coupled electro-mechanical finite elements analysis. The computations followed a standard finite element formulation procedure, in which the governing partial differential equations are approximated by a system of nonlinear algebraic equations and the resulting system of nonlinear equations solved using Newton–Raphson's iterative method (see the Methods for more details). For the mechanical constitutive equation, a neo-Hookean model[19] was used with a shear modulus of 312 kPa and Poisson ratio of 0.495 (nearly incompressible), based on measurements of the mechanical deformation response of the elastomer (see Methods for more details). The electrical constitutive equation is defined by a linear relation between the electric displacement and electric field with the relative permittivity of 5.5. Figure 2 compares the measured profile of a positive curvature dome with increasing electric field with the fully coupled electro-mechanical simulations.

**Dome-like shape with positive Gaussian curvature**. To realize shape morphing to a positive Gaussian curvature, an axisymmetric multilayer disk was fabricated consisting of 12 elastomer layers sandwiching 11 circular electrodes whose radii decrease linearly from 11 mm on the top layer to 1 mm on the bottom

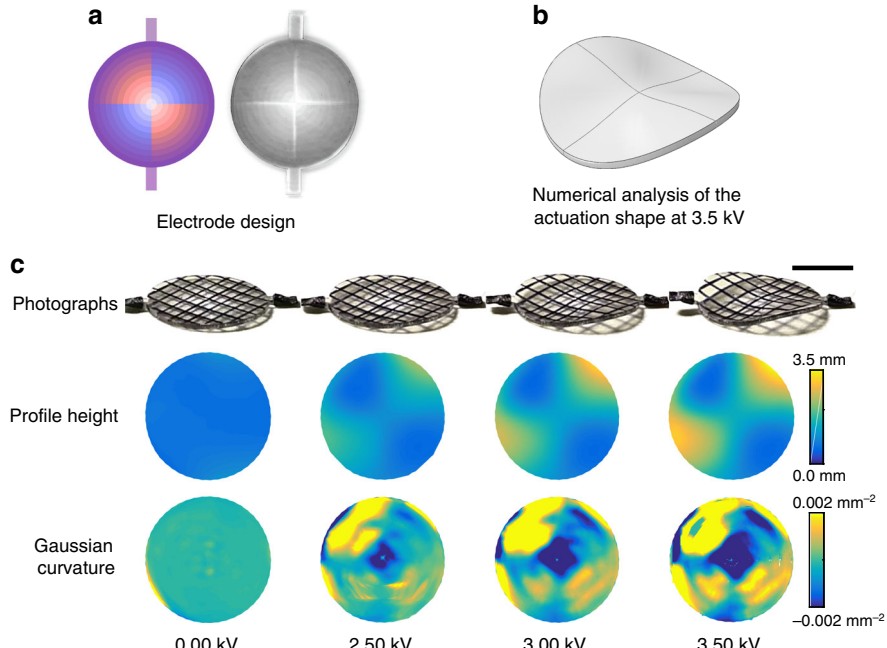

**Fig. 3** A thin sheet of elastomer morphing into a saddle with negative Gaussian curvature. **a** The design of the electrodes to produce a negative Gaussian curvature is such that there is less electrode overlap at the center of the elastomer sheet and increasing with radius, shown schematically with the color intensity on the left. Additionally, the electrodes are designed such that the blue quadrants bend outward and the red quadrants bend inward. On the right a photograph of the elastomer sheet is shown. **b** Numerical analysis of this structure shows that the flat sheet of elastomer morphs into a saddle shape upon applying a voltage. **c** A series of photographs of the disk with increasing voltage illustrates increasing negative curvature, confirmed by the height profiles and the Gaussian curvature derived from the 3D height data. Scale bar: 10 mm

layer. The resulting meso-architecture is characterized by the overlapping of the adjacent electrodes decreasing with radius, shown with the color intensity in Fig. 2a, but increasing linearly (in steps) with height through the thickness of the elastomer. Design of each individual electrode is shown in Supplementary Figure 1. The active regions are defined by the overlapping of these adjacent electrodes. Applying a voltage produces a differential actuation strain that decreases with radius and increases with height, and therefore the elastomer morphs into a dome shape, as predicted through the numerical analysis in Fig. 2b. Photographs and quantification of the shape change as a function of voltage, using a laser sheet light scanner, is presented in Fig. 2c, together with the Gaussian curvature computed from the heights at the indicated voltages. Real-time video recording of actuation of the thin sheet of elastomer into a dome shape is shown in Supplementary Movie 1. The measured height profiles and vertical displacement of the center of the disk as a function of voltage are compared with the predictions of the coupled electromechanical analysis in Fig. 2c. Excellent agreement is found between the measured profile and numerical simulations. It is also noteworthy that not only are the strains large, but the displacements can also be as large as several millimeters for a disk with 11 mm radius.

**Saddle-like shape with negative Gaussian curvature.** Changing the meso-architecture of the electrodes changes the Gaussian curvature produced when a voltage is applied. For instance, a thin flat disk of dielectric elastomer morphs into a saddle-like shape with negative Gaussian curvature when the actuation strain increases with radial position through the thickness of the elastomer. Such a strain field can be readily produced in a multilayer dielectric elastomer. In Fig. 3 this was accomplished using 11 circular electrodes sandwiched between 12 elastomer layers. The electrodes were arranged so that their overlap increases linearly

with radius, and with the additional feature that on two opposite quadrants of the disk it increases with height, while on the other two quadrants it decreases. This electrode configuration is represented by the blue and red color in the projected electrode density (Fig. 3a). Design of each individual electrode is shown in Supplementary Figure 2. Applying a voltage produces a strain field that increases radially, bending the elastomer sheet inwards on two opposite quadrants, and outwards on the other two opposite quadrants, resulting in a saddle actuation shape with negative Gaussian curvature, as predicted through numerical analysis (Fig. 3b) and shown in the experiment (Fig. 3c). The negative curvature increases with increasing voltage and when the voltage is removed the disk returns to its initial flat shape. Real-time video recording of actuation of the thin sheet of elastomer into a saddle shape is shown in the Supplementary Movie 2.

**Torus segments with positive and negative Gaussian curvature.** A third simple example is a strip that can morph into a torus segment with positive curvature (the outer part of a torus, Fig. 4a), or negative curvature (inner part of a torus, Fig. 4b). The number of elastomer and electrode layers was the same as in the previous examples, but both the electrodes and elastomer layers were rectangular and the same length (60 mm). The electric field induced curvature arises from the variation in width of the electrodes. For the torus segment with positive curvature on the left in Fig. 4a, the electrode width decreases from 11 mm on the bottom layer to 1 mm on the top layer. For the right one in Fig. 4a, the electrode width decreases from 22 to 2 mm. For the torus segment with negative curvature (Fig. 4b), the electrode on each layer consists of two rectangles of length 60 mm placed at the two edges of the elastomer sheet and their width increases from 11 mm on the bottom layer to 1 mm on the top layer. The design of the electrodes for the torus segments is shown in more details in Supplementary Figure 3.

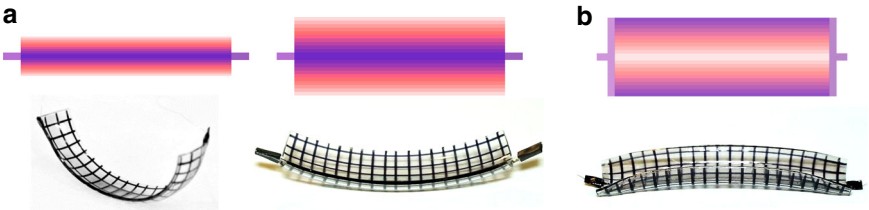

**Fig. 4** Illustration of morphing flat sheets of dielectric elastomer into torus segments with positive and negative Gaussian curvatures. The corresponding meso-architecture of the electrodes is shown on the top. **a** A rectangular sheet of elastomer morphs into a torus segments with positive Gaussian curvature when the electrodes are a set of rectangular strips that are 60 mm long and whose width changes linearly from 11 mm on the bottom layer to 1 mm on the top layer, shown on the left. For the other torus segment with positive Gaussian curvature that is shown on the right, the electrodes are a set of rectangular strips that are 60 mm long and whose width changes linearly from 22 mm on the bottom layer to 2 mm on the top layer. Scale bar: 20 mm. **b** A rectangular strip of elastomer morphs into a torus segment with negative Gaussian curvature when the electrodes are designed such that all adjacent electrodes overlap at the two edges of the strip, and the overlapping decreases to no overlapping at the center of the strip. Scale bar: 20 mm

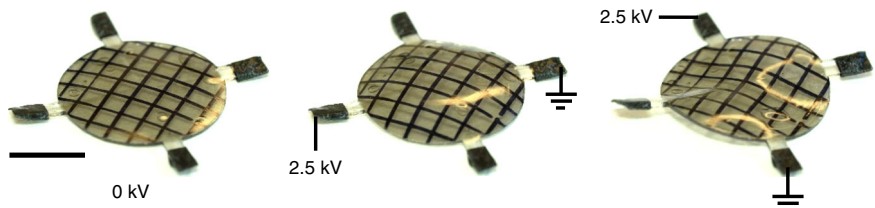

**Fig. 5** An initially flat thin circular sheet of elastomer (left) morphs into a dome shape (middle) and saddle shape (right), based on which sets of electrodes are addressed, illustrating a simple example of electrode-addressable reconfigurability. The contours of reflected light highlight the local curvatures. Scale bar: 10 mm

**Reconfigurable shape-morphing dielectric elastomers**. As shown, the multi-layering fabrication process enables a wide variety of electrode meso-architectures to be created. It also enables different sets of electrodes to be addressed and thereby produce morphing into different types of shapes according to which electrodes the voltage is applied to. Furthermore, it also enables different sets of electrodes to be powered with different voltages and at different time sequences enabling different shapes to be produced. As a result of this, for instance, in Fig. 5, two sets of addressable electrodes were incorporated into the elastomer sheet. One was a set of circular electrodes, as used to create a positive Gaussian curvature, and a second set of electrodes, as used for the saddle shape were positioned on alternate layers. To fabricate an elastomer disk with this electrode meso-structure, 23 layers of elastomer sandwiched 11 electrodes for the dome and 11 electrodes for the saddle. Supplementary Figure 4 illustrates the meso-architecture of the electrodes of the reconfigurable shape-morphing dielectric elastomer in details. Upon applying a voltage to the two sets of electrodes that correspond to the circular disk electrodes with varying radius, the elastomer morphs into a dome shape and applying voltage to the other set of electrodes results in a saddle shape (Fig. 5). Real-time video recording of the reconfigurable actuation is shown in Supplementary Movie 3.

## Discussion

Demonstration of the underlying concept of shape morphing in soft elastomers by manipulation of the spatial distribution of electric fields using an internal meso-architecture of electrodes lays the foundation for future device realization as well as more complex shaping. The possible shapes are fully deterministic as they depend on the three-dimensional architecture of the electrodes. Reconfigurable shape-morphing dielectric elastomers can lead to novel capabilities such as shape-morphing wings and bodies. In combination with internal power supplies and sensors, a variety of motion in response to a sensed signal can be envisaged. Finally, the ability to create complex shapes also poses a mathematical challenge, an inverse problem, of how to design the three-dimensional electrode arrangement to create specific targeted shapes.

## Methods

**Precursor**. The elastomer was made from an acrylic-based precursor:[18,20] 70% (by mass) of CN9018 (a urethane acrylate oligomer), 17.5% isodecyl acrylate (viscosity modifier), 5% isobornyl acrylate (toughness enhancer), 5% 1,6-hexanediol diacrylate (tunable crosslinker), 1% trimethylolpropane triacrylate (base crosslinker), 1% dimethoxy-2-phenylacetophenone, and 0.5% benzophenone (a photoinitiator).

Fifty gram batches of the ingredients were added together and mixed using a planetary centrifugal mixer (Thinky Mixer ARE-310) for 20 min at 2000rpm and defoamed for 30 s at 2200 rpm.

**Fabrication**. The fabrication process was similar to that used to make the multi-layer dielectric elastomer actuators:[18] the precursor is poured onto a wafer, spin coated for 2 min at 2000rpm, and then cured for 180 s under UV light (an array of 6 Hitachi F8T5-BL UV lamps with peak intensity at 366 nm wavelength) in the absence of oxygen, resulting in a 76 ± 2μm layer of elastomer.

A mask, defining the electrode's geometry, was placed onto the elastomer sheet and a mat of carbon nanotubes pressed on it to form a compliant electrode. The mat of carbon nanotubes was formed by vacuum filtration: a suspension of carbon nanotubes (P3-SWNT, Carbon Solutions, Inc.) in deionized water with 17% transmittance at 550 nm wavelength was prepared through sonication, centrifugation, and decanting[21]. Three hundred microliters of this suspension was vacuum filtered through a porous polytetra-fluorethylene (PTFE) filter membrane (0.2 μm pore and 47 mm diameter, Nuclepore®, Whatman, Florham Park, NJ, USA), resulting in mat of carbon nanotubes on the PTFE filter membrane with density of 6 mg m$^{-2}$ and sheet resistance of 6.7 ± 1 kΩ sq$^{-1}$ measured using a four-point probe sheet resistance measurement system (Keithley 6221 current source and Keithley 2182A Nanovoltmeter), and averaged over 22 measurements on 5 samples.

The procedure was continued by pouring the precursor onto the elastomer and repeating the spin coating, UV curing, and stamping steps for the desired number of electrodes.

**Mechanical characterization**. Uniaxial tension test was performed on a set of 14 dog-bone-shaped specimens with the geometrical parameters suggested by ASTM D412-16 (die C scaled by half). The specimens are stretched at a constant rate of 0.1 mm s$^{-1}$ until rupture and the tensile force was measured using a load cell (FUTEK LSB200, 2 lb, JR S-Beam Load Cell). To measure the stretch, the distance

between two marks, drawn on the narrow part of the dog-bone samples at a distance of 10 mm from each other, was measured as the specimen was stretched. The distance between these marks is measured from images recorded using a camera placed 30 cm above the sample. The measured distance over the initial distance gives the stretch. By fitting the Gent model[22] to the uniaxial stress-stretch curve, the shear modulus was determined $312 \pm 8$ kPa and the strain hardening parameter $J_{lim}$ was $4.3 \pm 0.3$, averaged over 12 measurements. Early strain hardening ($J_{lim} < 7$) prevents electro-mechanical instability. The stress-stretch curve and the fitted Gent model are shown in Supplementary Figure 5.

For the numerical analysis, however, the neo-Hookean model (with shear modulus of 312 kPa) is used instead of the Gent model. The two models agree well for the relatively small strains for which the strain hardening effect is negligible, that is, for the strains far from the rupture. Since the actuation strains in the dielectric elastomers studied in this paper are only a fraction of the ultimate strain, the neo-Hookean model can accurately describe the actuation of the dielectric elastomer sheets with less computational cost than the Gent model.

**Electrical characterization.** The electrical capacitance of the positive curvature actuator is measured using an LCR meter (Agilent E4980A) at 20 Hz in parallel circuit mode. The relative permittivity of the dielectric elastomer is calculated by comparing the measured capacitance to a COMSOL model of the actuator with relative permittivity of 1. Since capacitance is a linear function of permittivity, the ratio of the measured to the simulated capacitance gives the relative permittivity.

**Electrode design.** The meso-architecture of the electrodes for morphing into different shapes with positive and negative Gaussian curvatures is shown in Supplementary Figures 1–4. To morph into a dome-like shape with positive Gaussian curvature, Supplementary Figure 1, the meso-architecture consists of 11 concentric, inter-digitated circular electrodes whose radius decreases linearly from 11 mm on the bottom layer to 1 mm on the top layer. To morph into a saddle-like shape with negative Gaussian curvature, Supplementary Figure 2, the meso-architecture consists of 11 inter-digitated electrodes, where each electrode is a hollow circular disk whose outside radius is 11 mm and the inside radius changes linearly from 1 mm on the bottom layer to 11 mm on the top layer for two opposite quadrants and it changes linearly from 11 mm on the bottom layer to 1 mm on the top layer for the other two opposite quadrants. Therefore, the volume of the overlapping region increases linearly with radius, with the additional feature that on two opposite quadrants of the disk it increases with height, while on the other two quadrants it decreases.

Supplementary Figure 3 represents the meso-architectures of the electrodes to morph a flat strip of elastomer into torus segments with positive and negative curvatures. To morph into a torus segment with positive curvature the electrodes are a set of strips with constant length whose width decreases linearly from the bottom layer to the top layer. For the torus segment with negative curvature, the electrodes are designed such that all the adjacent electrodes overlap at the two edges of the strip and the overlap decreases to zero, linearly, at the center of the strip.

Supplementary Figure 4 shows the meso-architecture of the reconfigurable shape-morphing elastomer that morphs from a flat sheet into a dome-like shape with positive Gaussian curvature or a saddle-like shape with negative Gaussian curvature, based on which set of electrodes are addressed. The meso-architecture of the electrodes consists of alternating between the 11 electrodes of the dome-like shape and the 11 electrodes of the saddle-like shape.

**Profile measurement.** The three-dimensional profiles of the elastomer sheets were measured using a laser line scanner (MTI ProTrak PT-G 60-40-58) and a precision linear stage (GHC SLP35, GMC Hillstone Co. and MicroFlex e100 servo drive). The line scanner was mounted to the table and the elastomer sheet was placed onto the linear stage and the voltage ramped up in steps of 100V. In each step, the surface of the elastomer is scanned by moving the linear stage for course of 30 mm and speed of 3 mm s$^{-1}$, while the voltage is constant, resulting in about 25,000 points cm$^{-2}$ in 10 s.

**Numerical analysis.** Following the standard finite element formulation procedure[23,24], the governing partial differential equations are converted into a system of nonlinear algebraic equations. This system of nonlinear equations was then solved in Abaqus using Newton–Raphson's iterative method. An Abaqus user element (UEL) was developed to incorporate the coupling terms into the residual vector and stiffness matrix.

Actuation of dielectric elastomers is a nonlinear multi-physics problem, mathematically described by the balance of forces and the Gauss's flux theorem, together with a proper set of boundary conditions and constitutive equations for the material models. For quasi-static actuation, the balance of forces and Gauss's flux theorem are given by

$$\frac{\partial}{\partial x_i}\left(\sigma_{ij} + \sigma_{ij}^{Max}\right) = 0, \ j = 1, 2, 3,$$
$$\frac{\partial D_i}{\partial x_i} = q, \tag{1}$$

where $\sigma_{ij}$ is the Cauchy stress tensor, $\sigma_{ij}^{Max}$ is the Maxwell stress tensor, $D_i$ is electric displacement, and $q$ is the density of free charges. The two equations are coupled through the Maxwell stress tensor.

For the mechanical constitutive equation, a nearly incompressible neo-Hookean material model is adopted, which expresses the material's Helmholtz free energy, $\psi$, as a function of the first and the third invariants of the Cauchy–Green deformation tensors, $I_1$ and $I_3$:

$$\psi = \frac{\mu}{2}\left(\frac{I_1}{I_3^{1/3}} - 3\right) + \kappa\left(\sqrt{I_3} - 1\right)^2 = \frac{\mu}{2}(\bar{I}_1 - 3) + \kappa(J - 1)^2, \tag{2}$$

where $\mu$ is the shear modulus and equal to 312kPa, and $\kappa$ is the bulk modulus and is set to 32.2 MPa (two orders of magnitude higher than the shear modulus to represent a nearly incompressible material). For the neo-Hookean model that expressed the Helmholtz free energy as function of the first and the third invariants of the Cauchy–Green deformation tensors, the Cauchy stress is

$$\boldsymbol{\sigma} = \frac{2}{\sqrt{I_3}}\frac{\partial \psi}{\partial I_1}\mathbf{B} + 2\sqrt{I_3}\frac{\partial \psi}{\partial I_3}\mathbf{I}, \tag{3}$$

where $\mathbf{B}$ is the left Cauchy–Green deformation tensor and $\mathbf{I}$ is the Kronecker delta (identity matrix). For the dielectric material model, a linear polarization constitutive equation is used, where the electric displacement $D_i$ is a linear function of the electric field $E_i$:

$$D_i = \epsilon_0 \epsilon_r E_i, \ i = 1, 2, 3, \tag{4}$$

where $\varepsilon_0$, $\varepsilon_r$, and $E_i$ are vacuum permittivity, dielectric constant of the elastomer, and the components of electric field inside the elastomer. For an incompressible and linearly polarizable material, the Maxwell stress, $\sigma_{ij}^{Max}$, coupling the mechanical and electrostatic equations, is related to the electric field by

$$\sigma_{ij}^{Max} = \epsilon_0 \epsilon_r \left(E_i E_j - \frac{1}{2}E_k E_k \delta_{ij}\right). \tag{5}$$

The two governing partial differential equations (balance of forces and Gauss's flux theorem) along with the mechanical and electrical constitutive equations and the boundary conditions describes the actuation of dielectric elastomers. Following the standard finite element formulation procedure, this system of partial differential equations is written in integral form and integrated by part. A shape function $N^A$ is considered for the weight functions and the solution variables (displacement $u_i$ and electric potential $\phi$). The integration is performed using Gaussian quadrature. This procedure converts the set of four partial differential equations into the following system of $4N$ nonlinear algebraic equations, where $N$ is the total number of the nodes:

$$\left\{\begin{array}{c} R_{u_j}^A \\ R_\phi^A \end{array}\right\} = \left\{\begin{array}{c} -\sum_{n_G}\left(\sigma_{ij} + \sigma_{ij}^{Max}\right)\frac{\partial N^A}{\partial x_i}w_{n_G}\det\frac{\partial x_p}{\partial \xi_q} + \sum_{n_G}\left(t_j N^A\right)w_{n_G}\det\frac{\partial x_p}{\partial \xi_q} \\ -\sum_{n_G}\left(D_i\frac{\partial N^A}{\partial x_i} + qN^A\right)w_{n_G}\det\frac{\partial x_p}{\partial \xi_q} + \sum_{n_G}\left(q_s N^A\right)w_{n_G}\det\frac{\partial x_p}{\partial \xi_q} \end{array}\right\} = \left\{\begin{array}{c} 0 \\ 0 \end{array}\right\}, \tag{6}$$

For $j = 1,2,3$ and $A = 1, \ldots, N$. $R_{u_j}^A$ and $R_\phi^A$ are the residual values for the displacements and electric potential at node $A$, respectively. $n_G$ and $w_{n_G}$ are the Gaussian quadrature point index and the weight associated with it, respectively. $\xi_q$ are the local coordinates of the element. $t_j$ are the components of the surface traction and $q_s$ is the area density of surface charges. These nonlinear algebraic equations were solved iteratively using Newton–Raphson's method:

$$\left\{\begin{array}{c} R_{u_j}^A \\ R_\phi^A \end{array}\right\}^{i+1} = \left\{\begin{array}{c} R_{u_j}^A \\ R_\phi^A \end{array}\right\}^{i} + \frac{\partial}{\partial u_k^B}\left\{\begin{array}{c} R_{u_j}^A \\ R_\phi^A \end{array}\right\}^{i}\delta u_k^B + \frac{\partial}{\partial \phi^B}\left\{\begin{array}{c} R_{u_j}^A \\ R_\phi^A \end{array}\right\}^{i}\delta \phi^B = \left\{\begin{array}{c} 0 \\ 0 \end{array}\right\}, \tag{7}$$
$$j = 1, 2, 3, \ A = 1, \ldots, N,$$

where the superscripts $i + 1$ and $i$ are the iteration number. In each iteration, the following linear algebraic equations need to be solved:

$$\begin{bmatrix} K_{u_j^A u_k^B} & K_{u_j^A \phi^B} \\ K_{\phi^A u_k^B} & K_{\phi^A \phi^B} \end{bmatrix}\left\{\begin{array}{c} \delta u_k^B \\ \delta \phi^B \end{array}\right\} = \left\{\begin{array}{c} R_{u_j}^A \\ R_\phi^A \end{array}\right\}, \ j = 1, 2, 3, \ A = 1, \ldots, N, \tag{8}$$

$$K_{u_j^A u_k^B} = \frac{\partial R_{u_j}^A}{\partial u_k^B}, \ K_{u_j^A \phi^B} = \frac{\partial R_{u_j}^A}{\partial \phi^B}, \ K_{\phi^A u_k^B} = \frac{\partial R_\phi^A}{\partial u_k^B}, \ K_{\phi^A \phi^B} = \frac{\partial R_\phi^A}{\partial \phi^B}. \tag{9}$$

The solution variables $u_k^B$ and $\phi^B$ are obtained through iteratively solving this system of linear algebraic equations until the residual vector and change of the solution variables are within the convergence criterion.

Abaqus CAE is implemented to define the geometry and the boundary conditions, mesh the geometry, assemble the element stiffness matrices and the

residual vectors, solve the system of algebraic equations through Newton–Raphson's method, and visualize the results. An Abaqus UEL is developed to incorporate the coupling terms $K_{u_i^A \phi^B}$ and $K_{\phi^A u_k^B}$ into the element stiffness matrix, and the Maxwell stress term into the element residual vector.

**Code availability**. The custom computer codes utilized during the current study are available from the corresponding author on reasonable request.

## Data availability
All data generated or analyzed during this study are included in the published article and are available from the corresponding author on reasonable request.

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

## Acknowledgements
This work was supported by the MRSEC through the National Science Foundation grant DMR 14-20570. The authors are grateful to Dr. Mihai Duduta for his helpful comments and his assistance with preparing the materials.

## Author contributions
D.R.C. conceived and supervised the project. E.H. performed the experiments and the numerical calculations. Both authors contributed to the writing of the manuscript.

## Additional information

**Competing interests:** The authors declare no competing interests.

