## [Peer Review File · Nature Communications]

Reviewers' comments:

Reviewer #1 (Remarks to the Author):

This work deals with dielectric elastomers. In particular, electrically controllable morphing capabilities for dielectric elastomers are studied in this study. The underlying idea is to carefully place internal electrodes via a unique layer-by-layer fabrication method. Numerical analysis was performed to confirm the actuation of the proposed dielectric elastomers. The manuscript is concise and well written.

- 1) Page 2: "currently, actuators based..." - Possibly need to change?
- 2) Figure 4 is a good illustration.. However, it is obvious and thus may not be needed?
- 3) Page 16, there may be a need to add more information about "electrode design" as it is the major contribution for this study?
- 4) Abstract: All the shape changes are reversible when the voltage is removed? Need some data to justify this statement?
- 5) Any data for longevity of the actuators? There may be an issue for inter-digitated and layered electrodes (i.e de-lamination)?

Reviewer #2 (Remarks to the Author):

This manuscript presents an interesting work of changing the configuration of dielectric elastomer film via spatially distributed electric field. However, this manuscript is supposed to be revised based on the following comments before further consideration.

1. In Page 4, the deformed Gaussian curvature shape should be due to the gradient expansion along height rather than that along axial direction.
2. In the caption of Fig. 2(a), the quantity that decreased should be overlapping area of the active regions rather than "the number of active regions". In Fig. 2(d) (right), the numerical analysis results should be added for comparison.
3. Fig. 3(d) and (e) could be reorganized into a new figure since they are irrelevant to (a) to (c). The difference in the configurations of the two photographs in Fig. 3(d) is suggested to be further explained regarding the dimensions.
4. In the Mechanical Characterization part, the uniaxial tensile test data is fitted by the Gent model. But the fitted parameters are not used for simulation or in other parts of this manuscript. Such contents could be deleted if not necessary. Besides, it seems that ASTM D416-16 is not the standard for tensile test.
5. Some potential applications of this work are suggested to be described.
6. The design of electrodes and wire connection of the reconfigurable sheet are supposed to be illustrated in the Extended Data, like Extended Data Fig. 2 and Extended Data Fig .3.
7. Some revisions are required to improve this manuscript. This manuscript is sort of not well organized to me, especially the Introduction part, and some subheadings might be added. There are also some grammar mistakes and confusing expressions, like "It is also noteworthy that not only are the strains large, but the displacements can also be large, millimeters" in Page 7. The authors should check the whole manuscript carefully. Moreover, the equations are suggested to be numbered.
8. Dielectric elastomer is a typical soft material, which has the characteristics of instability under high electric field, which will lead to actuator failure. Reviewer suggests that author should add some articles about dielectric elastomer instability, to make this paper more comprehensive, such as:
 - 1) A nonlinear field theory of deformable dielectrics, J. Mech. Phys. Solids 56, 467 (2008).

- 2) Electromechanical stability of dielectric elastomer, *Appl. Phys. Lett.* 91, 061921 (2007).
- 3) Thermoelectromechanical stability of dielectric elastomers undergoing temperature variation. *Mechanics of Materials*, 2014, 72, 33-45.
- 4) Electromechanical instability and snap-through instability of dielectric elastomers undergoing polarization saturation. *Mechanics of Materials*, 2012, 55, 60-72.

Reviewer #3 (Remarks to the Author):

The paper addresses a very interesting application for active elastomers and proposes a technology based on varying the electric fields responsible for the activation of deformation by means of an innovative fabrication method.

Although the use and the performances of elastomers as active materials to produce large deformation is a well known outcome of at least twenty years of research study, the paper present a novel concept, based on a layer-by-layer fabrication of elastomers sheets and relevant electrodes.

As an hint to improve the references of this paper one might cite the following:
Carpi F et al 2008, *Dielectric elastomers as electromechanical transducers Fundamentals.*,
Materials, Devices, Models and Application of an Emerging Electroactive Polymer Technology.
Elsevier Science, Amsterdam.

The use of variable through-the-thickness induces strain is a very well known approach to drive bending deformation in a plate-like structure. The authors proposed a novel technology based on layering to generate variable strain along the thickness.

The paper is in fact well structured and presents very clearly the proposed methods, especially in the Section "Methods: precursor, fabrication, mechanical characterization, electrical characterization and Electrode design".

The section on numerical methods, especially where the finite element formulation is cited (referred as "standard") should be improved by stating in a more clear way the nonlinear and multiphysics nature of the problem and the consequent nonlinearity of the numerical approach and relevant simulations.

To this respect the authors may consider to refer to the following paper:
Lampamini I. et al "3D Finite Element Analyses of Multilayer Dielectric Elastomer Actuators with Metallic Compliant Electrodes for Space Applications, *Journal of Intelligent Material Systems and Structures*, Vol 21, April 2010, 621-632.

A complete and quite extensive presentation of experimental results complete the work.

This reviewer suggests tha author to proceed to the small suggested improvements and amendments; provided that minor corrections are done the paper is eligible for publication.

NCOMMS-18-28541-T. Response to Reviewer #1

This work deals with dielectric elastomers. In particular, electrically controllable morphing capabilities for dielectric elastomers are studied in this study. The underlying idea is to carefully place internal electrodes via a unique layer-by-layer fabrication method. Numerical analysis was performed to confirm the actuation of the proposed dielectric elastomers. The manuscript is concise and well written.

1) Page 2: "currently, actuators based...." - Possibly need to change?

Response: We have rephrased the paragraph to make it clearer and cited additional references (#15,16). Our claim "currently, actuators based on dielectric elastomers cannot morph in shape" is based on the argument that for a shape-morphing surface to morph from a flat sheet into a sophisticated shape, it must be able to change its Gaussian curvature (as the result of inhomogeneous deformation and hence inhomogeneous metric tensor), as discussed in the first and third paragraph. The Gaussian curvature is an intrinsic property of the surface and therefore if a surface morphs into a new shape, then it must change its Gaussian curvature from the first surface to the second surface. Current dielectric elastomer actuators provide in-plane or bending deformations, but the Gaussian curvature does not change under either of these. Under bending, for instance, as only one of the principal curvatures changes, Gaussian curvature which is the product of the two principal curvatures remains zero.

2) Figure 4 is a good illustration. However, it is obvious and thus may not be needed?

Response. Reconfigurability is one of the main attributes of using the electrode mesoarchitecture rather than any other shape-morphing based on differential swelling, differential stiffness, patterned liquid crystals, for instance. We use Figure 4 to illustrate this capability associated with addressability of different sets of electrodes since we believe that this makes the paper more interesting and fundamental as well as clearer to the readers.

3) Page 16, there may be a need to add more information about "electrode design" as it is the major contribution for this study?

We have expanded the description of the "electrode design" as suggested. This has been done in the "electrode design" section, the body of the article, and using the supplementary figures S2-S5.

4) Abstract: All the shape changes are reversible when the voltage is removed? Need some data to justify this statement?

Yes, indeed, all the shape changes are reversible when the voltage is removed. This is illustrated in the three supplementary videos.

5) Any data for longevity of the actuators? There may be an issue for inter-digitated and layered electrodes (i.e. de-lamination)?

We have no quantitative data as to the distribution in lives of the actuators we have produced since we have not undertaken reliability studies. However, we have not observed any delaminations in any of the multilayer elastomers we have investigated. In part, we attribute this to the bonding between successive elastomer layers through the gaps between the individual CNTs making up the percolative electrodes.

NCOMMS-18-28541-T. Response to Reviewer #2

This manuscript presents an interesting work of changing the configuration of dielectric elastomer film via spatially distributed electric field. However, this manuscript is supposed to be revised based on the following comments before further consideration.

1. *In Page 4, the deformed Gaussian curvature shape should be due to the gradient expansion along height rather than that along axial direction.*

Response: When there is only a gradient of expansion along the thickness direction (but the deformation is homogeneous along the surface of the sheet of elastomer), then the elastomer sheet simply bends along an axis. Under a simple bending along an axis, however, the Gaussian curvature remains zero as explained in the first paragraph and repeated in response to the first reviewer: The Gaussian curvature is the product of the two principal curvatures of the surface; under bending, one principal curvature changes and the other principal curvature, along the bending axis, remains zero, and therefore, the Gaussian curvature remains zero.

In fact, inhomogeneous deformation along the thickness is not even essential for changing the Gaussian curvature; for instance, a flat sheet of elastomer would morph into a dome shape even if the deformation is homogeneous along the thickness but inhomogeneous along the radius (larger expansion in the center of the sheet and reducing with radius). Examples of these can be found in the classical works we cite in the first paragraph, for instance the 4th reference (Klein, Y., Efrati, E. and Sharon, E., 2007. Shaping of elastic sheets by prescription of non-Euclidean metrics. *Science*, 315(5815), pp.1116-1120).

There are cases where an inhomogeneous deformation along the thickness can be used as an additional control and enhancement for shape-morphing. For instance, for the dome, one can choose whether the surface morphs into a concave dome or a convex dome, by imposing an inhomogeneous expansion along the thickness.

2. *In the caption of Fig. 2(a), the quantity that decreased should be overlapping area of the active regions rather than “the number of active regions”. In Fig. 2(d) (right), the numerical analysis results should be added for comparison.*

Following the reviewer’s comment, we modified the caption and we added the numerical analysis results to Fig. 2(d) (right).

3. *Fig. 3(d) and (e) could be reorganized into a new figure since they are irrelevant to (a) to (c). The difference in the configurations of the two photographs in Fig. 3(d) is suggested to be further explained regarding the dimensions.*

We agree and following the reviewer’s suggestion, we have divided figure 3 into two separate figures. A scale bar has been added to the figure of the torus segments as well as additional explanation on how the dimensions of the elastomer sheet affects the shape of the torus segment.

4. *In the Mechanical Characterization part, the uniaxial tensile test data is fitted by the Gent model. But the fitted parameters are not used for simulation or in other parts of this manuscript. Such contents could be deleted if not necessary. Besides, it seems that ASTM D416-16 is not the standard for tensile test.*

The uniaxial tension test was fitted by the Gent model and the shear modulus obtained from the fitting was used in our finite element modelling. The reason to choose the Gent model is that it can accurately describe the entire range of the stress-strain curve of the elastomer used in this paper, as shown in the supplementary figure S1. Fitting the Gent model to the uniaxial tension test outputs the shear modulus of the elastomer and the strain hardening parameter J_{\max} . The shear modulus obtained from the fitting is then used in our finite element modelling. However, for the relatively small strains that we are dealing with, the difference between the Gent model and the simpler neo-Hookean model is insignificant, and therefore we have used the neo-Hookean instead of the Gent model for the finite element analysis but the shear modulus used in the finite element modelling comes from the mechanical characterization.

Regarding the ASTM standard, the reviewer is correct. We meant to write ASTM D412-16 but mistyped it. It is corrected now.

5. *Some potential applications of this work are suggested to be described.*

Following the reviewer's suggestion, examples of the potential applications of this work are added to the last paragraph.

6. *The design of electrodes and wire connection of the reconfigurable sheet are supposed to be illustrated in the Extended Data, like Extended Data Fig. 2 and Extended Data Fig .3.*

We have added another extended figure describing the electrode design of the reconfigurable shape-morphing elastomer.

7. *Some revisions are required to improve this manuscript. This manuscript is sort of not well organized to me, especially the Introduction part, and some subheadings might be added. There are also some grammar mistakes and confusing expressions, like "It is also noteworthy that not only are the strains large, but the displacements can also be large, millimeters" in Page 7. The authors should check the whole manuscript carefully. Moreover, the equations are suggested to be numbered.*

We have numbered the equations as suggested but we do not agree that we made any grammatical mistakes or that the manuscript was not well organized. We paid close attention to the structure and the flow of the manuscript and we have proofread it several times. Nevertheless, following the reviewer's comment, we went through the manuscript again to make sure that it is well-organized.

8. *Dielectric elastomer is a typical soft material, which has the characteristics of instability under high electric field, which will lead to actuator failure. Reviewer suggests that author*

should add some articles about dielectric elastomer instability, to make this paper more comprehensive, such as:

- 1) *A nonlinear field theory of deformable dielectrics, J. Mech. Phys. Solids 56, 467 (2008).*
- 2) *Electromechanical stability of dielectric elastomer, Appl. Phys. Lett. 91, 061921 (2007).*
- 3) *Thermoelectromechanical stability of dielectric elastomers undergoing temperature variation. Mechanics of Materials, 2014, 72, 33-45.*
- 4) *Electromechanical instability and snap-through instability of dielectric elastomers undergoing polarization saturation. Mechanics of Materials, 2012, 55, 60-72.*

Response: We have seen no evidence of electromechanical instability described by the reviewer and detailed in the references he (or she) mentions. Those pieces of analysis are detailed and well founded, expanding on the original physical argument proposed in 1955 by Stark and Garton. It is well established that the electromechanical instability that leads to thinning, which in turn suddenly increases the electric field to exceed the electrical breakdown condition, is often seen in single layer dielectric elastomers with no constraint on either side *when* filamentary electrical breakdown from defects does not occur first. We have not seen this electromechanical instability in any of our multi-layered systems and always see breakdown from obvious defects, such as dirt particles. (As a result of our multiple spin coating and stamping steps, we find dust incorporation is unavoidable). We have also tried, as explained in “mechanical characterization” section, to always use elastomer compositions that have strain hardening given by $J_{\max} < 7$, a criterion from the electromechanical instability analyses that is prescriptive for avoiding electromechanical instability. [This strain hardening condition is given in, for instance figure 2(b) in reference #23 (Henann, D.L., Chester, S.A. and Bertoldi, K., 2013. Modeling of dielectric elastomers: Design of actuators and energy harvesting devices. Journal of the Mechanics and Physics of Solids, 61(10), pp.2047-2066.).

NCOMMS-18-28541-T. Response to Reviewer #3

The paper addresses a very interesting application for active elastomers and proposes a technology based on varying the electric fields responsible for the activation of deformation by means of an innovative fabrication method.

Although the use and the performances of elastomers as active materials to produce large deformation is a well-known outcome of at least twenty years of research study, the paper present a novel concept, based on a layer-by-layer fabrication of elastomers sheets and relevant electrodes.

As a hint to improve the references of this paper one might cite the following:

Carpi F et al 2008, Dielectric elastomers as electromechanical transducers Fundamentals., Materials, Devices, Models and Application of an Emerging Electroactive Polymer Technology. Elsevier Science, Amsterdam.

The use of variable through-the-thickness induces strain is a very well-known approach to drive bending deformation in a plate-like structure. The authors proposed a novel technology based on layering to generate variable strain along the thickness.

The paper is in fact well-structured and presents very clearly the proposed methods, especially in the Section "Methods: precursor, fabrication, mechanical characterization, electrical characterization and Electrode design".

The section on numerical methods, especially where the finite element formulation is cited (referred as "standard") should be improved by stating in a clearer way the nonlinear and multiphysics nature of the problem and the consequent nonlinearity of the numerical approach and relevant simulations.

To this respect the authors may consider referring to the following paper:

Lampamini I. et al "3D Finite Element Analyses of Multilayer Dielectric Elastomer Actuators with Metallic Compliant Electrodes for Space Applications, Journal of Intelligent Material Systems and Structures, Vol 21, April 2010, 621-632.

A complete and quite extensive presentation of experimental results complete the work.

This reviewer suggests the author to proceed to the small suggested improvements and amendments; provided that minor corrections are done the paper is eligible for publication.

We appreciate all the reviewer's comments. Following the suggestions, we have

- a) *cited the book "Dielectric elastomers as electromechanical transducers" and*
- b) *modified the numerical method section to emphasize the nonlinear and multi-physics nature of the problem. Also, we have cited the article "3D Finite Element Analyses of Multilayer Dielectric Elastomer Actuators with Metallic Compliant Electrodes for Space Applications" as suggested.*

REVIEWERS' COMMENTS:

Reviewer #1 (Remarks to the Author):

It looks like the authors have addressed the most of concerns/comments.

Reviewer #2 (Remarks to the Author):

The authors have addressed most of the previous questions. However, some improvements are still needed. This manuscript could be accepted after some minor revisions.

1. In the sentence "The resulting meso-architecture is characterized by the number of active regions decreasing with radius...", it should also be overlapping area of the active regions rather than "the number of active regions".
2. It is fine to use the shear modulus fitted by Gent model in neo Hookean model. But it should be noted where this value is used in Page 6 to prevent confusions of readers.
3. In Fig 4a, it seems that the directions of major curvature in the two photos are different. The electrode widths in these two structures are different. A simple discussion about the two different configurations could be given.
4. Although some expressions did not affect the readability of this manuscript, I still could not figure out why there is a word "millimeters" in the sentence "It is also noteworthy that not only are the strains large, but the displacements can also be large, millimeters." Besides, in the sentence "...morphs from a flat sheet into a dome-like shape with positive Gaussian curvature of a saddle-like shape with negative Gaussian curvature" of Page 17 and 18, it might be "dome-like shape with positive Gaussian curvature or a saddle-like shape with negative Gaussian curvature"?
5. The following is the reviewer's personal opinion on the introduction part. The revision could be dependent on the authors' decision. It seems that the paragraph following the paragraph "In this work..." also discussed about the related background and thus this paragraph might be put ahead the "In this work..." paragraph. Some contents in the paragraph "To morph" is similar to that in the paragraph "In this work..." These two paragraphs might be integrated into one single paragraph.

Reviewer #3 (Remarks to the Author):

The authors have carefully reviewed their manuscript according to the requests by the reviewers. Namely they have addressed the points raised by this reviewer who considers the amendments in line with his suggestions.

The paper addresses a very interesting application for active elastomers and proposes a technology based on varying the electric fields responsible for the activation of deformation by means of an innovative fabrication method.

The paper presents a novel concept, based on a layer by layer fabrication of elastomers sheets and relevant electrodes. The authors proposed a novel technology based on layering to generate variable strain along the thickness.

The paper is in fact well structured, and presents very clearly the state of the art and the proposed methods, especially in the Section "Methods: precursor, fabrication, mechanical characterization, electrical characterization and Electrode design".

The section on numerical methods, especially where the finite element formulation has been improved in the present version.

A complete and quite extensive presentation of experimental results complete the work.

This reviewer considers the paper eligible for publication.

NCOMMS-18-28541-T. Response to Reviewer #2

The authors have addressed most of the previous questions. However, some improvements are still needed. This manuscript could be accepted after some minor revisions.

1. *In the sentence “The resulting meso-architecture is characterized by the number of active regions decreasing with radius...”, it should also be overlapping area of the active regions rather than “the number of active regions”.*

Response: The wording of this sentence, in page 7, is modified accordingly: “The resulting meso-architecture is characterized by the overlapping of the adjacent electrodes decreasing with radius ...”

2. *It is fine to use the shear modulus fitted by Gent model in neo Hookean model. But it should be noted where this value is used in Page 6 to prevent confusions of readers.*

Response: We added a paragraph to the “Mechanical characterization” section in the Methods in page 16 to avoid any confusion about how the shear modulus obtained from the mechanical characterizations is used for the numerical analysis.

3. *In Fig 4a, it seems that the directions of major curvature in the two photos are different. The electrode widths in these two structures are different. A simple discussion about the two different configurations could be given.*

Response: Analysis of the direction of the major curvature of the torii and its dependency on the aspect ratio of the rectangular sheet of elastomer is in fact out of the scope of our current work and requires a thorough analysis.

4. *Although some expressions did not affect the readability of this manuscript, I still could not figure out why there is a word “millimeters” in the sentence “It is also noteworthy that not only are the strains large, but the displacements can also be large, millimeters.” Besides, in the sentence “...morphs from a flat sheet into a dome-like shape with positive Gaussian curvature of a saddle-like shape with negative Gaussian curvature” of Page 17 and 18, it might be “dome-like shape with positive Gaussian curvature or a saddle-like shape with negative Gaussian curvature”?*

Response: In the sentence with “millimeters” in page 7, we would like to convey the scale of the displacements for a disk of elastomer with radius of 11 mm. We slightly rephrased the sentence so that it will read better. Regarding the other sentence in page 18, “of” is a typo and should have been “or”, which we have corrected now.

5. *The following is the reviewer’s personal opinion on the introduction part. The revision could be dependent on the authors’ decision. It seems that the paragraph following the paragraph “In this work...” also discussed about the related background and thus this paragraph might be put ahead the “In this work...” paragraph. Some contents in the paragraph “To morph” is*

similar to that in the paragraph “In this work...” These two paragraphs might be integrated into one single paragraph.

Response: That’s a good suggestion. We have merged paragraphs 2 and 4 so that now the first two paragraph gives a background and literature review of the shape morphing mechanisms and dielectric elastomers, and the 3rd paragraph introduces the idea that is explored in this manuscript.